# Using High-Frequency PAR Measurements to Assess the Quality of the SIF Derived from Continuous Field Observations

Shuai Han [1,2], Zhigang Liu [1,2,*], Zhuang Chen [1,2], Hao Jiang [1,2], Shan Xu [1,2,3], Huarong Zhao [4,5] and Sanxue Ren [4,5]

1   State Key Laboratory of Remote Sensing Science, Jointly Sponsored by Beijing Normal University and Aerospace Information Research Institute of Chinese Academy of Sciences, Beijing Normal University, Beijing 100875, China; hstsinghua@mail.bnu.edu.cn (S.H.); 201921051173@mail.bnu.edu.cn (Z.C.); 202121051189@mail.bnu.edu.cn (H.J.); xushan@njau.edu.cn (S.X.)
2   Beijing Engineering Research Center for Global Land Remote Sensing Products, Institute of Remote Sensing Science and Engineering, Faculty of Geographical Science, Beijing Normal University, Beijing 100875, China
3   Plant Phenomics Research Centre, Academy for Advanced Interdisciplinary Studies, Nanjing Agricultural University, Nanjing 210095, China
4   Chinese Academy of Meteorological Sciences, Beijing 100081, China; 656892rzr@163.com (H.Z.); zhr680317@163.com (S.R.)
5   Hebei Gucheng Agricultural Meteorology National Observation and Research Station, Baoding 072656, China
*   Correspondence: zhigangliu@bnu.edu.cn; Tel.: +86-136-5105-1881

**Abstract:** Fluctuations in illumination are one of the major sources for SIF retrieval errors during temporal continuous field measurements. In this study, we propose a method for evaluating the quality of SIF based on simultaneous measurements of photosynthetically active radiation (PAR), which are acquired using a quantum sensor at a sampling frequency higher than that obtained using spectral measurements. The proposed method is based on the coefficient of variation (known as relative standard deviation) of the high-frequency PAR during a SIF measurement to determine the quality of the SIF value. To evaluate the method, spectral and PAR data of a healthy maize canopy were collected under various illumination conditions, including clear, cloudy, and rapidly fluctuating illumination. The SIF values were retrieved by 3FLD, SFM, and SVD. The results showed that SFM and 3FLD were sensitive to illumination stability. The determination coefficients ($R^2$) between PAR and SIF extracted by SFM and 3FLD were higher than 0.8 on clear and cloudy days and only approximately 0.48 on the day with rapidly fluctuating illumination. By removing the unqualified data using the proposed method, the $R^2$ values of SFM and 3FLD on the day of rapidly fluctuating illumination significantly increased to 0.72. SVD was insensitive to illumination stability. The $R^2$ values of SVD on days with clear, cloudy, and rapidly fluctuating illumination were 0.73, 0.76, and 0.61, respectively. By removing the unqualified data, the $R^2$ values of SVD were increased to 0.66 on the day with rapidly fluctuating illumination. The results indicated that the quality assessment method based on high-frequency PAR data can eliminate the incorrect SIFs due to unstable illumination. The method can be used to extract more accurate and reliable SIF datasets from long-term field observations for the study of the relationship between SIF and vegetation photosynthesis.

**Keywords:** SIF; high-frequency PAR; illumination stability; data quality

## 1. Introduction

Vegetation photosynthesis is a key driver of the global carbon cycle. Sun-induced chlorophyll fluorescence (SIF) is closely related to vegetation photosynthesis. Therefore, SIF is considered a promising remote sensing indicator to estimate gross primary production at the regional and global scales. Although, in recent years, SIF has been retrieved from airborne [1–4] and satellite-based remote sensing systems [5–7], ground-based measurements of SIF are essential to validate the results of airborne and satellite-based systems

and to study the relationship between SIF and vegetation photosynthesis [8]. Therefore, an increasing number of long-term automatic field observation systems of SIF have been set up around the world [9,10].

In ground-based observations, the majority of SIF inversion algorithms require not only the upwelling radiance spectrum ($L$) on top of the canopy, but also the simultaneous incident irradiance spectrum ($E$) at the surface [11]. Considering that a spectrometer can only receive the signals from a single optical channel at one time, two types of SIF observation systems, including the simultaneous system and the sequential system [12], have been developed. In a simultaneous system, two spectrometers measure $E$ and $L$ simultaneously [13]. As the performance of each spectrometer can change differently with time, temperature, humidity, and other factors [14], the two spectrometers must be intercalibrated frequently. Thus, simultaneous systems are not widely used in field measurements. In a sequential system, a spectrometer is connected to an optical path switcher to measure $E$ and $L$ in sequence [15]. Currently, sequential systems are widely used due to their ease of maintenance. To retrieve SIF accurately with the data acquired by sequential systems, the measured $E$ ($E_M$) should match the actual irradiance ($E_L$) during $L$ measurement. However, in field measurements, rapid changes in illumination can cause a temporal mismatch between $E$ and $L$ measurements. In other words, $E_M$ is different from $E_L$. During SIF retrieval in the field, temporal mismatch is one of the primary causes of errors [16].

To reduce the possibility of a temporal mismatch between $E$ and $L$, several methods have been proposed to shorten the measurement interval between $E$ and $L$, such as increasing the light flux of the fiber [17], improving the integration time optimization method [18], and improving the measurement process [9]. The above methods, however, cannot completely prevent the occurrence of a temporal mismatch when illumination rapidly changes in the field. To ensure the accuracy of the analyses, some researchers only use data collected under clear skies [15], which renders a great deal of field data useless.

Cogliati et al. [11] proposed a data quality indicator (named $DQ_S$) to account for the stability of $E$ during the sandwich measurement section, in which two irradiance spectra ($E1$ and $E2$) are acquired just before and immediately after the $L$ measurement. $DQ_S$ is calculated as the percentage difference between $E1$ and $E2$. $DQ_S$ is currently widely used to filter out abnormal data caused by temporal mismatches between $E$ and $L$. However, this indicator does not account for variations in illumination between irradiance measurements, which may also cause inversion errors.

If a quantum sensor is added to the SIF measurement system, PAR values can be recorded at a higher frequency than that of E measured by the spectrometer in the system, enabling a more accurate evaluation of illumination stability. Therefore, in this study, we use a SIF observation system equipped with a quantum sensor to perform observation experiments on a maize canopy. Our research objectives are to: (1) develop methods for SIF quality assessment and data filtering using high-frequency PAR data; (2) assess the effectiveness of our method under different sky conditions; and (3) evaluate the applicability of our method to different SIF extraction algorithms.

## 2. Methodology

### 2.1. Field Ground Measurements

#### 2.1.1. Observation System

In this study, we used an improved AutoSIF (Bergsun Inc., Beijing, China) to observe the SIF and PAR of a maize canopy automatically. The observation system includes a control computer, two spectrometers, an optical multiplexer, and a photon meter (Figure 1). The two spectrometers include a QE65pro spectrometer (Ocean Optics Inc., Dunedin, FL, USA) and an HR4000 spectrometer (Ocean Optics Inc., Dunedin, FL, USA). The QE65pro spectrometer has a spectral range of approximately 640 nm to 800 nm, a high spectral resolution (SR) of approximately 0.3 nm, and a spectral sampling interval (SSI) of approximately 0.15 nm. Its signal-to-noise ratio (SNR) can reach approximately 1000:1. The HR spectrome-

ter has a spectral range of 198 nm to 982 nm and a SR of approximately 1 nm. To reduce dark current and noise, the two spectrometers were contained within a refrigeration system chamber that maintains a temperature below 25 °C. Since the QEpro spectrometer had a higher RS and SNR, we used only the data collected by the QEpro spectrometer to retrieve SIF in this study. The optical multiplexer (MPM-2000, Ocean Optics Inc., Dunedin, FL, USA) had 16 different observation channels, and the 2 spectrometers were connected to 8 channels. Two of the channels were used to measure the incident irradiance for the two spectrometers, two channels to measure the dark current, and the other channels to measure the upwelling radiance of the different targets. The observation sequence of the AutoSIF was to first measure the target with a QE spectrometer and then immediately measure the same target with an HR spectrometer. In one measurement section of a target, a spectrometer performed the following steps: (1) integration time optimization of the *E*, (2) measurement of the *E*, (3) measurement of the dark current, (4) integration time optimization of the *L*, (5) measurement of the *L*, and (6) measurement of the dark current. An up looking fiber with a CC-3 cosine-corrector probe (Ocean Insights Inc., Dunedin, FL, USA) was used to measure *E*. A down looking bare optical fiber with a FOV of 25° was used to measure the *L* of the canopy. The PAR data were collected using a quantum sensor (ML-020P, EKO Instruments Co., Ltd., Tokyo, Japan). The quantum sensor was set to sample every 250 milliseconds. Approximately two to four PARs could be recorded in a second due to data loss. Since the spectrometer data and the photon meter data were collected by the same computer, the timing of the two data was matched.

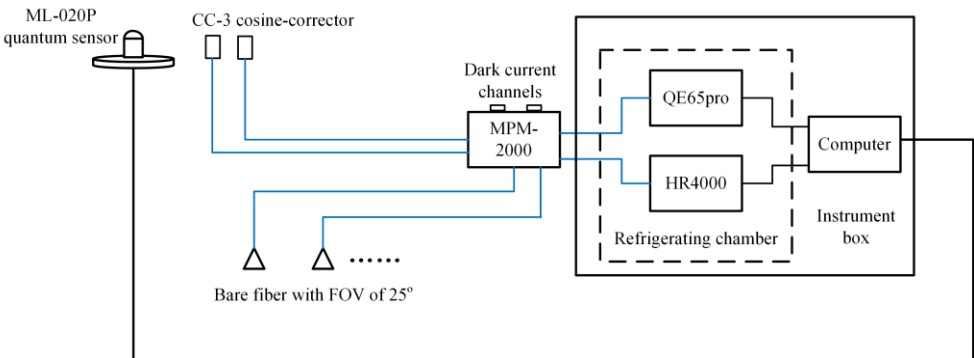

**Figure 1.** Schematic illustration of the improved AutoSIF structure. The solid blue lines represent the fibers. The black connecting lines represent the data transmission lines.

2.1.2. Measurement Scheme

The field experiments were conducted in the city of Baoding, in the province of Hebei, China (39.14455°N, 115.73785°E) (Figure 2). There were six plots of the same size of 2 m × 4 m. Five of the plots were planted with maize, and only one of the maize plots was properly irrigated. In this study, the relationship between the retrieved SIF and PAR was used to evaluate the quality of SIF since the true value of canopy SIF is difficult to obtain [19]. However, the relationship between canopy SIF and PAR may not be linear under water stress [20,21]. Therefore, only the data from the plot without water stress were used in this study.

or the measurements of the QE65pro spectrometer in the studied plot, one bare fiber was mounted at a distance of approximately 2 m above the maize canopy to measure the L from the nadir. The ground instantaneous field of view of bare fiber is a circle with a diameter of about 0.89 m. A CC-3 cosine-corrector probe was installed vertically at the same height as the canopy to measure incident irradiance. To obtain the matching incident solar radiation for the quantum sensor, it was mounted at the same height as the CC-3 cosine-corrector probe, keeping the sensor horizontal and free of shadow interference. Before the experiment, we performed radiometric and wavelength calibrations. The observations of the canopy were conducted on every no-rain day between 8:00 and 16:00 between DOY

214 and DOY 234 in 2020. In total, the observation data of 17 days were collected. In DOY 214-234, the maize was between the jointing stage and flowering stage. In addition, the LAI of the canopy of maize was about 3–3.5 in DOY 224-230.

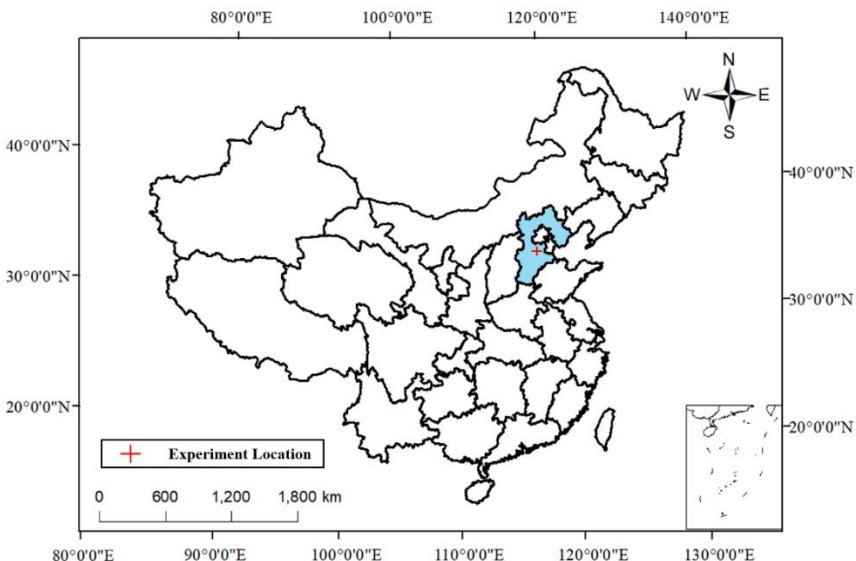

**Figure 2.** The location of the experiment site.

### 2.2. SIF Retrieval

In this study, we applied three frequently applied methods for the proximal sensing of SIF [8,22], including modified Fraunhofer line discrimination (3FLD) [23], the spectral fitting method (SFM) [24,25], and the singular-vector-decomposition-based method (SVD) [19,26,27].

The 3FLD method assumes a linear variation of reflectance ($\rho$) and fluorescence ($F$) over the absorption window and uses the following formula to calculate the SIF value at the band $\lambda_{in}$ in the absorption feature:

$$F = \frac{L(\lambda_{in}) - E(\lambda_{in}) \cdot \frac{\omega_l L(\lambda_l) + \omega_r L(\lambda_r)}{\omega_l E(\lambda_l) + \omega_r E(\lambda_r)}}{1 - \frac{E(\lambda_{in})}{\omega_l E(\lambda_l) + \omega_r E(\lambda_r)}} \tag{1}$$

where $L$ is the upwelling radiance; $E$ is the incident irradiance; $\lambda_l$ and $\lambda_r$ represent the two reference bands situated on the left and right shoulders of the absorption feature, respectively; and $\omega_l$ and $\omega_r$ are the weighting factors. In this study, we set $\lambda_l$, $\lambda_{in}$, and $\lambda_r$ as 759.0, 760.0, and 767.0 nm, respectively, to retrieve SIF at the $O_2A$ absorption band.

The SFM method [25] uses general mathematical functions representing canopy $\rho$ and $F$ within narrow spectral windows centered at absorption features. SFM uses mathematical formulations to describe the spectral shapes of true reflectance ($\rho$) and fluorescence ($F$).

$$L(\lambda) = E \cdot \rho_{mod}(\lambda) + F_{mod}(\lambda) + \epsilon(\lambda) \tag{2}$$

where $\rho_{mod}$ represents the mathematical formulation to describe the spectral shape of $\rho$; $F_{mod}$ represents the mathematical formulation to describe the spectral shape of $F$; and $\epsilon$ is the modeling error. In this study, we used quadratic polynomials to model $\rho$ and $F$ with a band range of 759.0–767.0 nm.

The SVD method is a data-driven statistical approach based upon the assumption that the SIF-free spectrum can be represented by a linear combination of singular vectors derived from the singular vector decomposition of a given SIF-free training dataset [26,27].

$$L(\lambda) = \left(\sum_{i=0}^{np} a_i \cdot \lambda^i\right) \cdot \left(\sum_{j=1}^{nv} b_j \cdot SV_j\right) + F_S \cdot h_F \cdot \tau^\uparrow \tag{3}$$

where $\lambda$ represents the wavelength; the polynomial of $\lambda$ represents spectrally smooth terms, such as surface reflectance and atmospheric scattering; np is the order of the polynomial; *SV* is the singular vector obtained by singular value decomposition from the training dataset to describe the variability in solar irradiance and atmospheric transmittance; *nv* is the number of *SV*s describing the high spectral frequency variations; $h_F$ is the fluorescence shape function; $F_S$ is the scaling factor of $h_F$; $a_i$ and $b_j$ represent the coefficients for polynomial and singular vectors, respectively; and $\tau^\uparrow$ is an upwelling atmospheric transmittance. In this study, we used the algorithm proposed by Chang [19]. We set the band range to 750.0–765.0 nm and used the daily irradiance spectra as the training dataset.

### 2.3. Methods for Evaluating Data Quality and Selecting Data

Since the sampling frequency of the quantum sensor was higher than that of the spectrometer, multiple *PAR* data could always be collected during the measurements of *E* and *L*. We calculated the coefficients of variation of incident *PAR*, including $CV_E$, $CV_L$, and $CV_{section}$, to evaluate illumination variations during the measurement intervals. $CV_E$, $CV_L$, and $CV_{section}$ represent the illumination variations during the irradiance measurement, the radiance measurement, and the measurement section, respectively (Figure 3). They were calculated as follows:

$$CV_E = \frac{\sigma_{(PAR_E)}}{\mu_{(PAR_E)}} \tag{4}$$

where $\sigma_{(PAR_E)}$ represents the standard deviation and $\mu_{(PAR_E)}$ represents the mean *PAR* recorded during the downwelling irradiance measurement.

$$CV_L = \frac{\sigma_{(PAR_L)}}{\mu_{(PAR_L)}} \tag{5}$$

where $\sigma_{(PAR_L)}$ represents the standard deviation and $\mu_{(PAR_L)}$ represents the mean *PAR* recorded during the upwelling irradiance measurement.

$$CV_{section} = \frac{\sigma_{(PAR_{section})}}{\mu_{(PAR_{section})}} \tag{6}$$

where $\sigma_{(PAR_{section})}$ represents the standard deviation and $\mu_{(PAR_{section})}$ represents the mean *PAR* recorded during the measurement section.

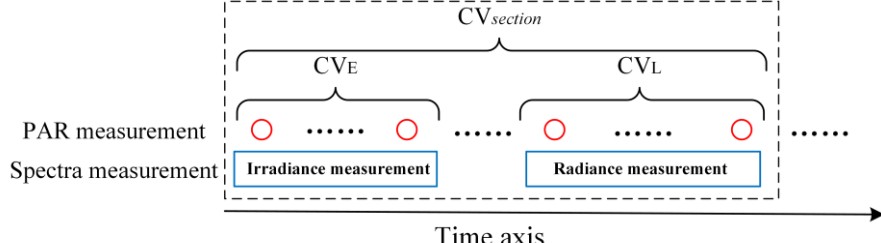

**Figure 3.** The calculations of $CV_E$, $CV_L$, and $CV_{section}$. The bottom axis represents time. The red cycles represent the high-frequency *PAR* measurements of the ML-020P quantum sensor. The blue rectangles represent the spectral measurements of the spectrometer, including the measurements of radiance and irradiance.

Using the three *CV*s, the following rules were applied to filter out low-quality SIF data retrieved by the SFM and 3FLD.

$$CV_E < a \ \& \ CV_L < b \ \& \ CV_{section} < c \tag{7}$$

where *a*, *b*, and *c* are the thresholds. Since the SVD method computes the SIF value solely based on the *L* measurement, only illumination stability during *L* measurement should be considered. Therefore, we used a rule based on $CV_L$ to remove low-quality SIF data retrieved by SVD.

$$CV_L < b \tag{8}$$

The number of quantum sensor measurements used for the calculation of the three *CV*s depends on the length of the spectral measurement period. In this experiment, the average number of quantum sensor measurements used to calculate $CV_E$, $CV_L$, and $CV_{section}$ were about 4, 6, and 48, respectively.

As mentioned in Section 2.1.2, the relationship between the retrieved SIF and PAR was used to evaluate the performance of the above data filtering methods.

## 3. Results

### 3.1. The CVs of PAR during Different Sky Conditions

The diurnal dynamics of high-frequency PAR and the three *CV*s during three typical days are shown in Figures 4 and 5. On DOY 224, the sky was clear for most of the day, and the diurnal variation curve of PAR was close to the cosine curve, with only a few sharp fluctuations (Figure 4a). On this day, most (~99%) of the $CV_E$ and $CV_L$ values (Figure 5a,b) were lower than 0.5%. The $CV_{section}$ values were generally greater than the values of $CV_E$ and $CV_L$. However, most (~90%) of the $CV_{section}$ values were still smaller than 0.5% (Figure 5c). On DOY 227, the illumination fluctuated rapidly and frequently (Figure 4b). On this day, the number of data points with *CV* values greater than 0.5% was significantly higher than that on DOY 224 (Figure 5d–f). On DOY 230, the sky was cloudy during most of the day, and the PAR was lower than that on the other days (Figure 4c). On this day, all the $CV_E$ and $CV_L$ values (Figure 5g,h) were lower than 0.5%, and most (~55%) of the $CV_{section}$ values were still smaller than 0.5% (Figure 5i). The CVs of high-frequency PAR data can accurately depict the stability of the illumination during spectral measurements.

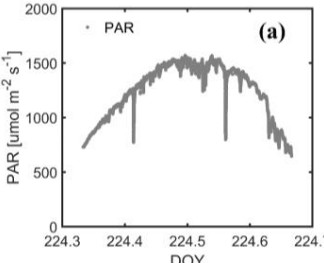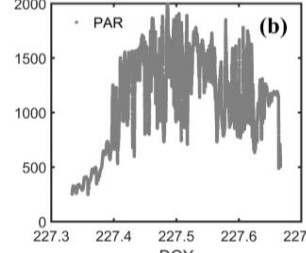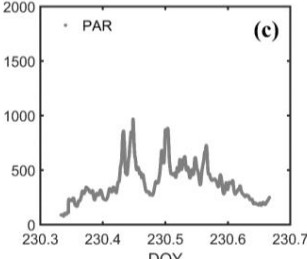

**Figure 4.** The diurnal dynamics of PAR under three sky condition days: (**a**) the clear day (DOY 224), (**b**) the rapidly fluctuating illumination condition day (DOY 227), and (**c**) the cloudy day (DOY 230). The gray dots represent the high−frequency PAR data.

### 3.2. Performance of Data Quality Assessment and Data Selection

According to the range of the CVs on the clear day (Figure 5a–c), we set the value of a, b, and c in the filter rules Equation (7) for 3FLD and SFM as 0.5%. The value of b in the filter rule Equation (8) for SVD was also set as 0.5%. Figure 6 shows the dynamics of SIF retrievals of three days under different illumination conditions.

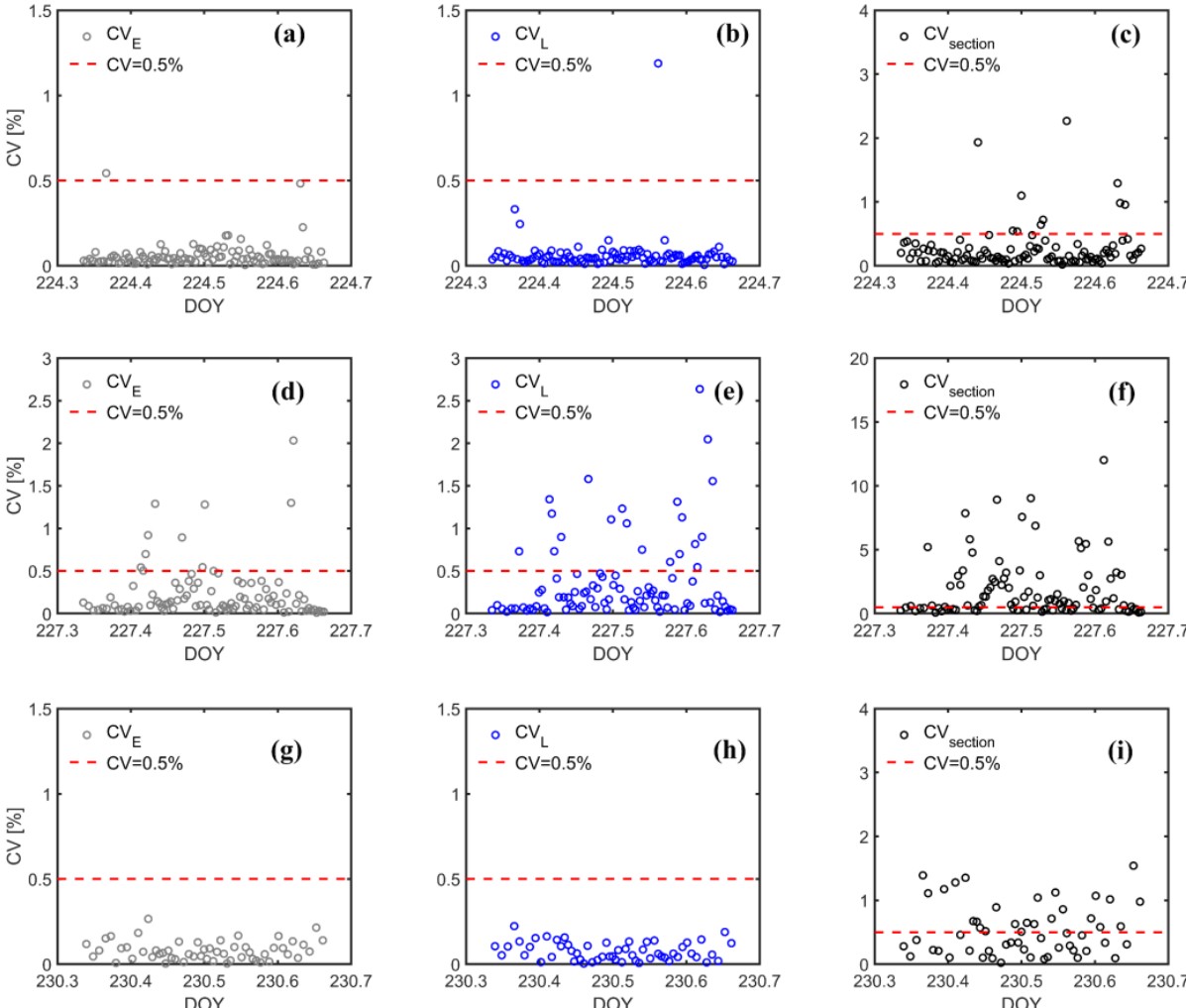

**Figure 5.** The diurnal dynamics of $CV_E$, $CV_L$, and $CV_{section}$ under three sky condition days: (**a–c**) for the clear day (DOY 224); (**d–f**) for the rapidly fluctuating illumination condition day (DOY 227); and (**g–i**) for the cloudy day (DOY 230). The gray dots represent the $CV$s of downwelling irradiance measurements. The blue dots represent the $CV$s of upwelling radiance measurements. The black dots represent the $CV$s of the measurement sections.

　　　DOY 224 was mostly clear throughout the day. There were only a few data (~12%) that were filtered out for 3FLD, SFM, and SVD (Figure 6a–c). The $R^2$ values between the PAR and SIF values obtained by all three SIF retrieval methods were all higher than 0.7 and did not change significantly after data filtering (Figure 7a–c).

　　　On DOY 227, the illumination fluctuated rapidly. This resulted in many outliers and even negative SIF values retrieved using the 3FLD and SFM algorithms (Figure 6d,e). The $R^2$ values between the SIF and PAR values were less than 0.5. After data filtering, these $R^2$ values increased significantly to approximately 0.72 (Figure 7d,e). However, the SVD results were very different on this day from the 3FLD and SFM results (Figures 6f and 7f). Before data filtering, the $R^2$ value of SVD was approximately 0.6, which was obviously higher than those of 3FLD and SFM. After data filtering, the $R^2$ value of SVD only increased by a small margin to 0.67. This is primarily because on this day, only 23% of the data was filtered out for the SVD method, which was significantly lower than the proportions of 3FLD and SFM (~65%).

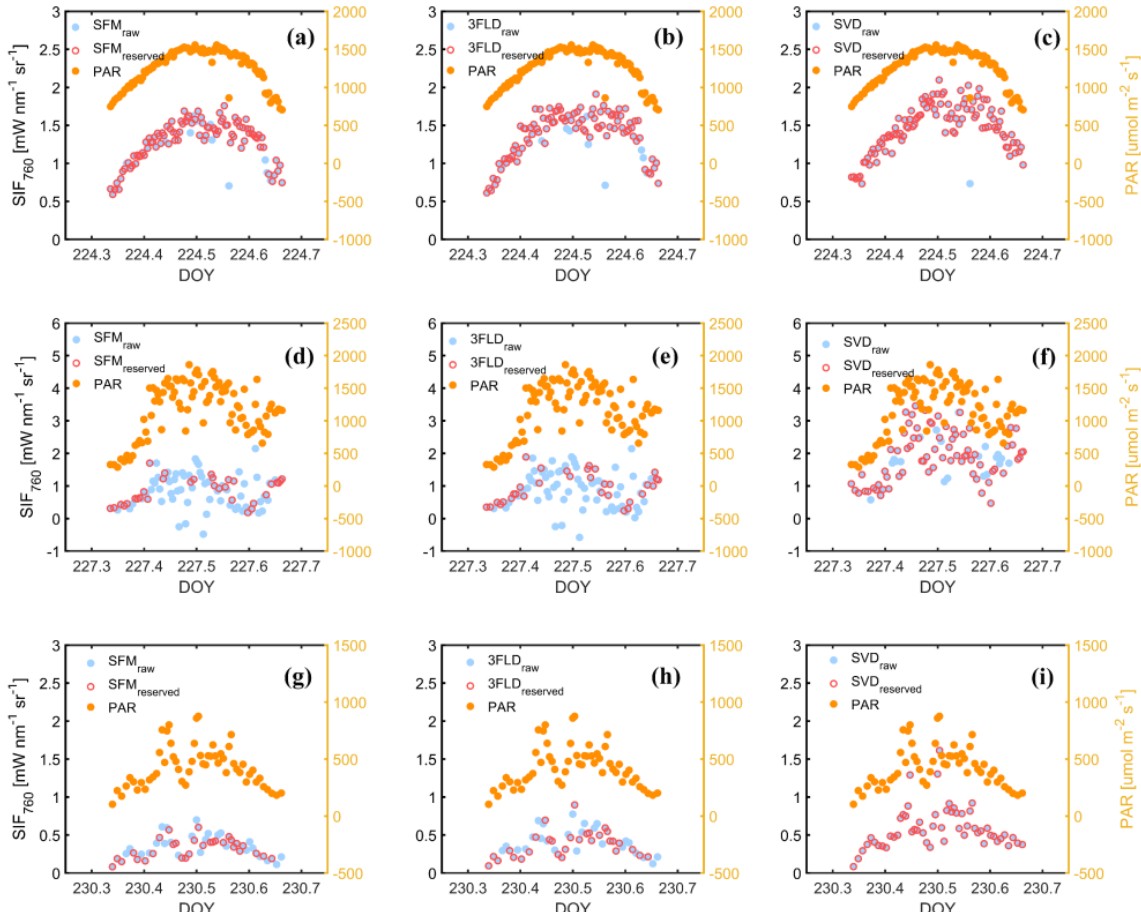

**Figure 6.** The diurnal dynamics of SIF retrieved by the SFM, 3FLD, and SVD methods and the diurnal dynamics of PAR under different sky conditions, such as (**a–c**) for the clear day (DOY 224); (**d–f**) for the rapidly fluctuating illumination condition day (DOY 227); and (**g–i**) for the cloudy day (DOY 230). The blue dots represent the retrieved SIF at 760 nm by the three algorithms. The red circles represent the reserved SIF. The orange dots illustrate the average values of the high–frequency PAR data during the $L$ measurements.

DOY 230 was cloudy. For most of the day, the light intensity fluctuated more slowly than it did on DOY 227 (Figure 4c). Therefore, the proportions of the filtered data of 3FLD and SFM on this day were lower than the proportions of DOY 227 (Figure 6g,h). In addition, SVD data were entirely retained (Figure 6i). After filtering the data, the $R^2$ values of the 3FLD and SFM algorithms increased from ~0.83 to ~0.93 (Figure 7g,h). The $R^2$ of SVD was ~0.76 (Figure 7i), which was lower than the values of 3FLD and SFM.

We also utilized all observation data on the 17 days without rain from the measurement period (DOY 214-234) to verify the proposed method. After eliminating the data with unstable illumination conditions, the $R^2$ values of the 3FLD and SFM algorithms increased from ~0.64 to ~0.72 (Figure 8a,b). However, the $R^2$ of SVD was slightly improved (Figure 8c) and lower than the $R^2$ of 3FLD or SFM. These results show that the data quality assessment method is significantly effective for the 3FLD and SFM algorithms and slightly effective for the SVD algorithm.

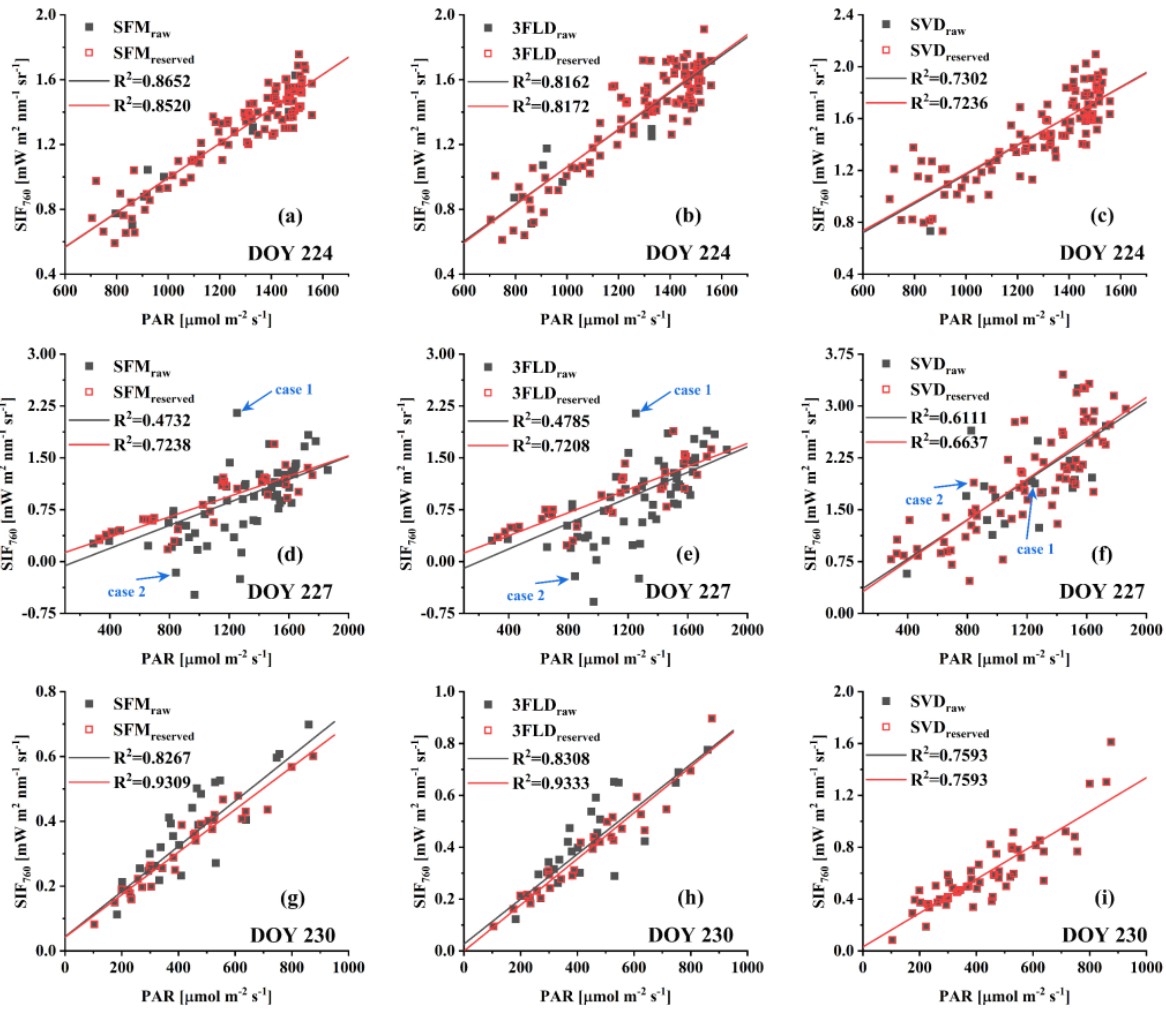

**Figure 7.** The relationship between *PAR* and SIF retrieved by the SFM, 3FLD, and SVD methods under three sky conditions: (**a–c**) for the clear day (DOY 224); (**d–f**) for the rapidly fluctuating illumination condition day (DOY 227); and (**g–i**) for the cloudy day (DOY 230). Black square dots represent the retrieved SIFs at 760 nm by SFM, 3FLD, and SVD. Red square circles represent the reserved SIF. The *PAR* values were the average of the high-frequency *PAR* values during the *L* measurements.

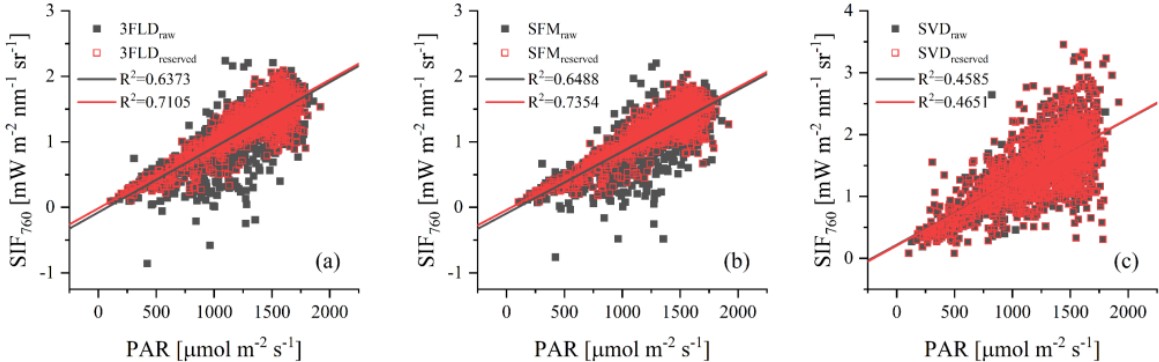

**Figure 8.** The relationship between PAR and SIF retrieved by the (**a**) SFM, (**b**) 3FLD, and (**c**) SVD methods from 17 days across the measurement period. Black square dots represent the retrieved SIF at 760 nm by SFM, 3FLD, and SVD. Red square circles represent the reserved SIF. The PAR values were the average of the high-frequency PAR values during the L measurements.

To demonstrate how illumination affects SIF retrieval accuracy, we examined two typical cases. In the first case, the mean of the *PAR* values during the measurement of the $E1$ spectrum ($PAR_{E1}$) was approximately 75% of the mean of the PAR values during the measurement of the $L$ spectrum ($PAR_L$) (Figure 9a). This indicates that the real $E$ was stronger during the measurement of the $L$ spectrum. When the measured $E$ was assumed to be correct, the fluorescence fill bulge of the apparent reflectivity curve in the absorption band increased (Figure 9b), and the SIF values retrieved by 3FLD and SFM were overestimated (case 1 in Figure 7d,e). In contrast, in the second case, the mean of the $PAR_{E1}$ values was greater than the mean of the $PAR_L$ values (Figure 9c). Assuming the measured E to be true, the apparent reflectivity curve appeared concave (Figure 9d) rather than convex at the $O_2A$ absorption band. The SIF values retrieved by 3FLD and SFM were underestimated (case 2 in Figure 7d,e). In both cases, however, the SIF values retrieved by SVD were not abnormal.

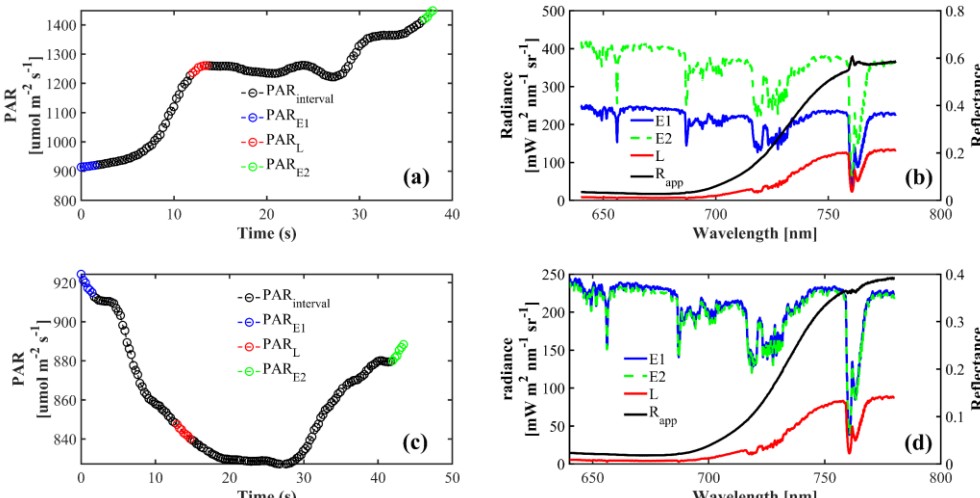

**Figure 9.** The PAR and spectral data during two measurement cases. (**a**,**b**) for the case 1 and (**c**,**d**) for the case 2. The red dots, blue dots, and green dots represent the PAR measured during the upwelling radiance measurement ($L$) and two irradiance measurements ($E1$ and $E2$) acquired just before and immediately after the $L$ measurement, respectively. The blue solid line, green dotted line, and red solid line represent the spectra of $E1$, $E2$, and $L$, respectively. The black solid line represents the reflectance ($R_{app}$) of the maize canopy calculated by $E1$ and $L$.

## 4. Discussion

For the sequential system of the SIF measurement, which is widely used in field automatic measurements of SIF, the difference between the measured $E$ and the actual $E$ during the measurement of $L$ is one of the primary reasons for the error in the estimation of SIF [8,15]. In this study, we added a *PAR* sensor to the SIF observation system to continuously measure irradiances at a frequency significantly higher than spectral measurements. Based on the coefficients of variation calculated by high-frequency *PAR* values, we proposed a method to evaluate the illumination stability during each SIF measurement to select reliable SIF measurements. Our results show that the correlations between *PAR* and SIF of the 3FLD and SFM methods were significantly reduced when the illumination changed rapidly, which is consistent with the existing research [19]. The proposed method could effectively delete abnormal data caused by rapid illumination fluctuation and significantly improve the correlation between the PAR and reserved SIF values (Figures 7 and 8). Chang et al. [19] proposed using the coefficients of variation of incident PAR recorded by a quantum sensor and a clearness index to classify sky conditions. The results of this study suggested that the use of high-frequency PAR data can be further used to evaluate the data quality of SIF.

When compared to the commonly used data quality indicator $DQ_S$ [11], the method proposed in this study has a higher accuracy when the illumination changes rapidly. As an example, although the magnitudes of the E values before and after the L measurement in case 2 were comparable, the magnitude was significantly greater than the actual value during the L measurement (Figure 9c,d). $DQ_S$ cannot identify this abnormal observation, but the method proposed in this study can. $DQ_S$ has the advantage of not requiring an additional quantum sensor. However, adding a quantum sensor to a SIF observation system is simple and inexpensive. Furthermore, some SIF observation systems and quantum sensors are integrated with eddy covariance systems [9]. Therefore, the PAR observation data can be conveniently used for quality assessment.

Our results show that, without data filtering, the applicability of SVD to various illumination conditions was better than that of SFM and 3FLD (Figure 7f). The results are consistent with those of Chang et al. [19]. This is because the SVD does not suffer from temporal mismatches between L and E, since it estimates E through a training dataset as opposed to using a special measure of E. However, even for the SVD, removing the error data based on high-frequency PAR could improve the estimation accuracy when the illumination was rapidly fluctuating (Figure 7f). After removing the error data, the correlation between PAR and SIF estimated by the SVD was lower than that of the SFM and 3FLD (Figures 7 and 8). This may be due to the accuracy of the SVD algorithm itself. This could also be related to the parameter settings or the training dataset selection [28,29]. How to select the optimal parameters and training dataset for SVD remains a challenging problem. In addition, we found that the SIFs estimated by the SVD were generally higher than those estimated by the SFM and 3FLD (Figure 6). When the illumination changed rapidly, the phenomenon became more pronounced (Figure 6d–f). This phenomenon is consistent with previous studies [30]. The cause of the phenomenon is still unknown.

We found that, when the PAR during the measurement of L was lower than the PAR during the measurement of E, the apparent reflectance curve collapsed in the $O_2$-A absorption band, leading to an underestimation of SIF or even a negative value (Figure 9c,d). This may be due to the change in the ratio of direct to diffuse radiation in the incident irradiance. When the solar altitude angle is the same, the ratio under strong illumination is generally higher than that under weak illumination. Although the effect of the ratio of direct to diffuse radiation on SIF extraction has been studied [16,31], it remains to be explored how the difference between the ratio during the E measurement and the ratio during the L measurement may affect the extraction of SIF.

## 5. Conclusions

When field measurements of SIF are conducted continuously over an extended period, large and unpredictable fluctuations in illumination may result in estimation errors. To address this issue, we proposed in this study to make use of high-frequency PAR data to assess illumination stability, and then used these data to assess the quality of SIF data. The experimental results of a healthy maize canopy indicated that, by using the proposed method to eliminate low-quality data, the qualities of the SIF retrieved by the 3FLD, SFM, and SVD algorithms were improved. Under cloudy and rapidly changing illumination conditions, the correlations between PAR and SIF retrieved by 3FLD and SFM were significantly improved after low-quality data were excluded. Therefore, using the data quality assessment method proposed in this paper, reliable SIF data can be selected from the observation results under various illumination conditions, which is crucial to correctly interpret SIF dynamics, detect stresses, establish mechanistic relationships between SIF and GPP, and validate satellite SIF products.

**Author Contributions:** Conceptualization, S.H. and Z.L.; methodology, S.H.; software, S.H.; formal analysis, S.H.; investigation, S.H. and Z.L.; resources, S.H., Z.C., H.J., S.X., H.Z. and S.R.; writing— original draft preparation, S.H.; writing—review and editing, Z.L.; visualization, S.H.; supervision, Z.L.; project administration, Z.L.; funding acquisition, Z.L. All authors have read and agreed to the published version of the manuscript.

**Funding:** This research was funded by the National Natural Science Foundation of China (grant number 42071402) and the Water Resources Science and Technology Project of Jiangxi Province, China (202023ZDKT10, KT201706).

**Acknowledgments:** We thank the Natural Science Foundation of China for their support.

**Conflicts of Interest:** The authors declare no conflict of interest.

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
