# Peer review of "Using High-Frequency PAR Measurements to Assess the Quality of the SIF Derived from Continuous Field Observations"

_remotesensing, doi:10.3390/rs14092083_

Round 1

Reviewer 1 Report

This work touches problematics of error estimation in fluorescence estimation using equations applied to measured spectral curves. Some work regarding sensitivity of SIF measurements in fluctuating light conditions was presented already by Damm et al (around 2015, it may be reference 16), but another insight is for sure necessary. Then we see that it is possible, that different methods to estimate SIF differ in sensitivity, the difference may be largely influenced by light conditions. During my read of the manuscript I was facing some troubles, which are specified in the comments. The largest weakeness is the built of the abstract at the start, and methods description. Introduction provide good background, discussion is not somehow wrong. Details are written in comments.

Line 15: High‐quality field measurements of sun-induced chlorophyll fluorescence (SIF) under all illumination conditions are vital for interpreting SIF dynamics.  -  I do not see first introductory sentence in the abstract as very fortunate, because it is necessary to distinguish between relationship between fluorescence and physiology and fluorescence and impact of measuring technique on measured fluorescence. This sentence mix both impact already in the start, but not sufficiently distinguished. It may be not problem to start with the second sentence.

Line 20: Term coefficient of variation is not quite common, me personally see it for the first time, so I would prefer rather describe it in the abstract differently or avoid it. At first sight it looks to ma as ratio between frequencies. And at the end of abstract I get to the fact, that it may be yield of fluorescence emission calculated based on the incoming PAR light. So I would avoid mixing two terms, and focus on just one.

It may be better to describe proposed correction methods, it is also difficult to say what unqualified data may represent. Maybe better avoid this in abstract.

What was the reference to compare fluorescence estimated with SVD matrix.

Line 31: connection “illumination rapidly fluctuating day” may not be appropriate

Introduction provides sufficient background and clarify purpose of the article, but abstract introduces mess at the start, therefore it should be simplified.

Line 94: There is se of HR spectrometers, there is for sure HR2000, HR4000; I see from the brief insight into manual that HR4000CG-UV-NIR has resolution of 1nm, however it should be better indicated in text which type of spectrometer was used.   /  QEPro is standard, in Europe Flox system is commertialy developed and widely used, how AutoSIF instrument stand in comparison with this product, is it one of a kind instrument? Does it contain two pairs of spectrometers or two spectrometers of one kind? / Why is HR mentioned at all, if the data from it are not included later?

Line 103: “had 16 different observation channels, and the two spectrometers were connected to 8 channels” – I am little confused only QE was used to measure fluorescence, but 16 channels were used, in my counts each channel should measured at certain configuration, what we cannot avoid without changing background or rotating fibre, so I am confused how the instrument work, if just two units and fibres are used. This may require photograph of the machine or relevant reference. From information in row 105 I can imagine optical fibres are rotating, switching on and off to measure dark-current, but still reference spectrum may require cosine corrector or reference (spectralon) panel at least. An in row 112 I recognize again the probably the instrument work in dual field of view configuration. These information should be unified, but largest mess originate from information that 16 channels convert to 8 channels, but only 2 spectrometers are used.. and… HR is of minor importance.

Line 117: but the delay looks to be within second so negligible. But from the figure 7 I would say, the spectra were  taken randomly, because 15-25 seconds delay is quite long.

Line 130: So it should be indicated directly that “only the data from the plot without water stress were used in this study.” And minimize the rest of the information.

Line 138: “radiation calibrated “ is wrong connection

According to chapter 2.2 SIF were estimated manually performing counting. But were there any steps done before to ensure that the spectra (input data) were measured with high precission. Then spectra are measured at resolution 0.3 nm, and result is presented at 1 nm precision, what is the conversion here?

Line 191: But are not the two instruments supposed to measure simultaneously? The only one CV would be enough. The delay in the spectrometers between scans, if the operation switches must be hard to measure.

Line 202: OPTIONAL - but the measurements using spectrometers are mostly very quick within microseconds, regardless how many scans are put into average. And that may be mismatch between PAR sensor and spectrometer, or am I counting wrong? Accounted signal from PAR sensor would cover longer interval than that from spectrometer, by my consideration. But it may be not big mistake, we are still somehow describing shifts in light conditions. On the other hand, longevity of this interval, may be large determinant for observed error. So it is most likely question to Figure 2.

Line 250: I am not familiar with the “filtering” procedure. Did you just remove the data that did not fulfill some criteria, or the CV values have been used somehow to correct the SIF value? In other publication authors for example state, that SIF estimated from iFLD and SVD show almost no difference (Acebron 2021). But this manuscript is showing that fluctuating light can introduce errors in SIF retrieval. Why is SVD so superb, if the data were not filtered and showed good agreement with PAR. This is answered in discussion around line 325. And from another perspective of view, isn´t it possible that also another factors affect fluorescence in light conditions transitions, because it is quite known that SIF behaves differently in high light and low light conditions, there is dual response. Or is it just estimation error originating from formula, as is suggested around line 280. It is also known that older sFLD introduces mistakes, due to mathematical build of the formula. Most interesting information regarding underestimation and overestimations of SIF appears in this paragraph, but it is presented as of minor importance at start, but the key features somehow appear in the discussion. Key determinants in this work appear to be time delay.

Line 323: eddy covariance systems and PAR sensors are mostly not dependent.

Line 340: rather lower than less

Reviewer 2 Report

Dear authors,

thank you for interesting reading. I appreciate the simplicity of your design which should be very effective at the same time. I believe you will be able to conduct and publish more extensive and robust evaluation and comparison with existing methods during your forthcoming research.

You can find my comments/suggestions in the following text.

P1L25 - The correlation coefficient is usually denoted as "R". Based on the values in the following text I expect you meant coefficient of determination.
P2L68 - The reference No.18 appears as text not as an active object.
P4L140 - You describe the period of the year when you conducted the measurements. Please, consider also shortly describing the growth stage and canopy (eg. canopy cover, LAI). It could help other authors when comparing with your results.
P5 chapter 3.1 - Please, double-check all cross-references to figures and point to proper ones.
P6L240 - The cross-reference should probably point to Fig. 4a-c, when your text is about CV.
P6L242 - The sentence where you refer to Fig. 5 is vague. Please, consider reformulation to something like "Fig. 5 shows dynamics of SIF retrievals ..."
P7L255-257 - in the sentence where you describe the proportions. It is not 100 % clear whether 23% was filtered out or 23% was filtered as valid data. Please, refolmulate.
Figures 5 and 6 - Please, consider column titles depicting the retrieval method. Than, you don't have to write this information into each subplot.
                - Please, unify the marking of the data that were filtered as valid in figure title/description and subplot legend. You call them "selected" in the figure title/description and "reserved" in the legend. Try to be consistent when you use "names" of some information.
P8L279-281 - Please, check whether the PAR_E1 and the PAR_L should be used vice versa. From Fig. 7a I understand that the PAR_E1 is ~ 75% of the PAR_L.
P9L315-316 - You say, that your method has higher accuracy compared to DQs. By design it could/should provide better separation of measurements with high irradiance variability, but I kind of miss some direct comparison in your Results - some figure/table/text showing what is the difference in number of sections filtered out or in R2, when you apply the DQs to you dataset.
P9L323 - You say that the PAR from the eddy covariance systems can be used for your method. Could you provide some information on the minimal frequency of the PAR measurements that is necessary for your method?

Reviewer 3 Report

The manuscript proposed to use the simultaneous measurement of PAR for evaluating the quality of the SIF measurement, taking advantage of the high frequency of PAR measurement. The results showed the proposed method can eliminate the incorrect SIF due to unstable illumination. The topic is interesting and suitable for the Remote Sensing Journal. However, I have two concerns about this manuscript. 

1. The proposed method can only be applied to the SIF measurement with simultaneous PAR measurement. Relying on the simultaneous PAR measurement, the application situation of the proposed method is quite limited. While previous method (Cogliati et al. 2015) does not rely on other measurements. The comments of ‘this indicator do not account for variations in illumination between irradiance measurements, which may also cause inversion errors’ is not fair. Only the same illumination variation during E1 and E2 measurement will lead to quality indicator invalid, which rarely happens.   

2. Since the SIF estimated from the SVD method is not sensitive to the illumination variation, why not directly use the SIF estimated from the SVD method? The improvement of the proposed method to SVD derived SIF is quite limited (R2 from 0.61 to 0.66).

Minor: L22, L129-L131 The proposed method is mainly based on the variation of the illumination influence PAR; it is nothing to do with the status of the canopy. 

Reviewer 4 Report

See attached file. 

Round 2

Reviewer 4 Report

authors improved the manuscript according to my suggestions. Therefore, I think that now the paper is ready for publication.